# Therapeutic Potential of Ex Vivo Expanded γδ T Cells against Osteosarcoma Cells

**DOI:** 10.3390/cells11142164

**Published:** 2022-07-11

**Authors:** Yunmi Ko, Yeon Ho Jeong, Jun Ah Lee

**Affiliations:** 1Center for Pediatric Cancer, Department of Pediatrics, National Cancer Center, Goyang 10408, Korea; 75302@ncc.re.kr; 2Department of Medical Biotechnology, College of Biomedical Science, Kangwon National University, Chuncheon 24341, Korea

**Keywords:** chemo-immunotherapy, chemotherapeutic agents, immunotherapy, mononuclear cells, osteosarcoma, peripheral blood, γδ T cells

## Abstract

Immunotherapy is an attractive therapeutic strategy for the treatment of osteosarcoma (OS). The unique features of γδ T cells have made them popular for cancer immunotherapy. Here, we expanded γδ T cells using human peripheral blood mononuclear cells (PBMCs) and investigated their therapeutic potential against OS cells. PBMCs from healthy donors were cultured for 10 days with CON medium (unstimulated control); EX media, CON with recombinant human interleukin-2 (rhIL-2) and zoledronate; and EX28 media, CON with rhIL-2, zoledronate, and CD3/CD28 activator. The expanded γδ T cells were isolated by magnetic cell separation or fluorescence-activated cell sorting, cultured with two OS cell lines (KHOS/NP and MG-63) at various cell ratios with or without doxorubicin or ifosfamide, and analyzed for cytotoxicity and cytokine secretion. The number of CD3^+^γδTCR^+^Vγ9^+^ triple-positive γδ T cells and concentrations of IFN-γ and TNF-α were highest in the rhIL-2 (100 IU) and zoledronate (1 μM) supplemented culture conditions. The CD3/CD28 agonist did not show any additional effects on γδ T cell expansion. The expanded γδ T cells exhibited potent in vitro cytotoxicity against OS in a ratio- and time-dependent manner. The γδ T cells may enhance the effect of chemotherapeutic agents against OS and may be a new treatment strategy, including chemo-immunotherapy, for OS.

## 1. Introduction

Osteosarcoma (OS) is the most common malignant bone tumor, with a peak incidence in adolescents. Multimodal treatment comprising surgery and chemotherapy improves the survival of patients with localized OS of the extremities [1,2]. However, patients with metastatic or recurrent disease still have a poor prognosis [3,4,5]. Currently, treatment options are limited to patients with relapsed or refractory (R/R) OS, with few clinical trials.

Immune checkpoint inhibitors prolong the survival of many patients with cancer. However, agents targeting PD-1/PD-L1 or CTAL-4 did not demonstrate clear benefits in patients with R/R OS [6,7,8]. Immunotherapy is an attractive therapeutic strategy for the treatment of OS [9,10]. Cooperative clinical study groups have tested the efficacy of various immunotherapies, including muramyl tripeptide-phosphatidylethanolamine and interferon, for patients with OS. However, immunotherapy for OS is challenging [11,12,13]. Chimeric antigen receptor T cell-based therapeutic approaches have been precluded because of the absence of OS-specific antigens. In addition, macrophages, not T cells, constitute the largest proportion of immune cells infiltrating OS. Therefore, a new approach is required to develop immunotherapies for OS.

γδ T cells are a unique population of lymphocytes with functional profiles of both innate and adaptive immune properties. In contrast to the features of αβ T cells, γδ T cells can recognize diverse antigens without human leukocyte antigen restriction [14,15,16]. Recognition of phosphoantigens, such as those from *Mycobacterium tuberculosis*, allows γδ T cells to develop potent immune responses [17,18]. Moreover, γδ T cells produce cytokines, such as interferon-γ (IFN-γ) and tumor necrosis factor-α (TNF-α), and exert direct cytotoxicity in response to malignancies [19,20]. Despite the fact that γδ T cells constitute a small portion of the overall T cell pool, due to these unique features, they have become attractive targets for cancer immunotherapy.

In this study, we investigated the therapeutic potential of γδ T cells, focusing on the following questions:(1)What is the optimal ex vivo condition for expanding γδ T cells obtained from human peripheral blood?(2)Do ex vivo expanded γδ T cells show a better function (especially antitumor activity) compared to naïve γδ T cells?(3)Do ex vivo expanded γδ T cells have a synergistic effect with conventional chemotherapeutic agents against OS cells?

## 2. Materials and Methods

### 2.1. Expansion of Human γδ T Cells

This study was approved by the Institutional Review Board of the National Cancer Center (NCC2019-0236). After informed consent, peripheral blood mononuclear cells (PBMCs) (approximately 4–50 mL) were obtained from 15 healthy men aged 20–40 years. PBMCs were isolated using SepMate tubes (StemCell Technologies, Vancouver, Canada) and Ficoll-Hypaque (GE Healthcare, Chicago, IL, USA) according to the manufacturer’s instructions and were subsequently exposed to different culture conditions. PBMCs were suspended at 5 × 10^5^ cells/mL in T cell expansion media supplemented with 2% immune cell serum replacement and 200 mM L-glutamine (CON media; all from Thermo Fisher Scientific, Waltham, MA, USA) and exposed to different culture conditions, as follows:(1)CON media (unstimulated control);(2)EX media, CON media with 100 IU rhIL-2 and 1 μM zoledronate (EX media);(3)EX28 media, CON media with 100 IU rhIL-2, 1 μM zoledronate, and 25 µL/mL CD3/CD28 activator (StemCell Technologies, Vancouver, Canada).

After 10 days, the expanded cells were counted using an automated cell counter (Bio-Rad Laboratories, Hercules, CA, USA) and re-suspended in an appropriate medium or buffer for downstream applications according to the manufacturer’s instructions. The expanded γδ T cells were isolated by magnetic cell separation using a TCR γ/δ^+^ T Cell Isolation Kit (Bio-Rad Laboratories, Hercules, CA, USA) or by fluorescence-activated cell sorting (FACS).

### 2.2. Cytotoxicity and Cytokine Production of Expanded γδ T Cells

KHOS/NP and MG-63 OS cells were cultured in complete α-MEM supplemented with 10% fetal bovine serum (all from Thermo Fisher Scientific, Waltham, MA, USA), and co-cultured with expanded γδ T cells at different cell ratios with or without doxorubicin (DOX) or ifosfamide (IFO). DOX (LC Laboratories, Woburn, MA, USA) and IFO (Selleck Chemicals LLC, Houston, TX, USA) were diluted in culture medium, each with various concentrations: DOX, 0.0625–2 µM; and IFO, 1250–40,000 µM. After removing the old media containing γδ T cells, the viability of adherent OS cells was evaluated using the EZ-CYTOX assay kit (DoGenBio, Seoul, Korea) after 24, 48, and 72 h. Cytokine (IFN-γ and TNF-α) levels in the co-culture supernatants were measured using ready-to-use ELISA kits (R&D Systems, Minneapolis, MN, USA), according to the manufacturer’s instructions.

### 2.3. Flow Cytometry Analysis

Expanded γδ T cells were isolated by FACS using two or three surface markers (human CD3^+^γδTCR^+^ double-positive or CD3^+^γδTCR^+^Vγ9^+^ triple-positive). For FACS analysis, the cells were stained with antibodies against the following: anti-human CD3, CD4, CD8, αβTCR, γδTCR, and Vγ9 (all from BD Biosciences, San Jose, CA, USA). Flow cytometry was performed on a BD FACSCanto II and FACSVerse, and ten thousand to a million events were acquired per sample and analyzed using FACS Diva and FlowJo software (v.10; accessed on December 2017) (all from BD Biosciences, San Jose, CA, USA).

### 2.4. Statistical Analysis

All experiments were conducted in triplicate at a minimum. Statistical analysis was performed using GraphPad Prism software (v.9.3.1, GraphPad Software Inc., San Diego, CA, USA; accessed on 2 December 2021), and the significance levels were * *p* < 0.05, ** *p* < 0.01, and *** *p* < 0.001. Combination index (CI) analysis was performed using CalcuSyn software (v.2.1, Biosoft, Cambridge, UK; accessed on 3 December 2018), and CI values indicated synergism (CI < 1), additive effect (CI = 1), and antagonism (CI > 1) between expanded γδ T cells and anti-cancer drugs.

## 3. Results

### 3.1. Optimum Culture Conditions for Expansion and Activation of Human γδ T Cells

Paired samples of PBMCs obtained from the same donor were cultured in three different media (CON, EX, and EX28 media). The degree of γδ T cell expansion was compared first between CON and EX media (with hIL-2 and zoledronate), followed by EX and EX28 media. After 10-day culture in EX media, the total cell count increased over 50 times, which was significantly higher than that in CON media (4.3 × 10^5^ cells/mL in CON media and 3.3 × 10^7^ cells/mL in EX media, *p* < 0.0001) (Figure 1a). In human PBMCs, γδTCR^+^ γδ T cells represented less than 5% of CD3^+^ T cells; after 10-day culture in EX media, the proportion of γδTCR^+^ γδ T cells reached 80% of CD3^+^ T cells (4.4% in COM media and 75.4% in EX media, *p* < 0.0001) (Figure 1b). In contrast to the expansion of CD3^+^γδTCR^+^ γδ T cells, the proportion of CD3^+^αβTCR^+^ αβ T cells and their subsets (CD3^+^CD4^+^ and CD3^+^CD8^+^ T cells) decreased after 10-day culture in EX media (Figure 2a).

To investigate whether CD28 is required for the expansion and activation of γδ T cells, CD3/CD28 activator, designed to activate and expand human T cells, was added to EX media (EX28 media). As shown in Figure 1a, the total cell count was significantly higher in EX28 than in EX media after 10-day culture (1.94 × 10^8^ cells/mL vs. 1.67 × 10^7^ cells/mL; *p* < 0.0001). However, the proportion of CD3^+^γδTCR^+^ γδ T cells was significantly lower in EX28 media than in EX media (0.08% vs. 52.5%), showing that CD3/CD28 did not exert any effect on the expansion of human γδ T cells (Figure 1b). Then, we evaluated the expression levels of phenotypic markers of activation and cytokine production of γδ T cells under optimum culture conditions, with the addition of zoledronate (1 μM) and rhIL-2 (100 IU) in CON media. After 10-day culture in EX media, the proportion of the Vγ9^+^ cells reached more than 90 % of CD3^+^γδTCR^+^ double-positive T cells, and the supernatant concentrations of both IFN-γ and TNF-α increased compared with those in CON media (Figure 2b,c).

These results indicate that γδ T cells were successfully expanded and functionally activated in EX media with rhIL-2 and zoledronate. CD3/CD28 co-stimulation showed no effect on the expansion of γδ T cells; therefore, expansion culture conditions were optimized with zoledronate (1 μM) and rhIL-2 (100 IU), without a CD3/CD28 T cell activator.

### 3.2. Cytotoxicity of Ex Vivo Expanded γδ T Cells against OS Cells

After PBMCs were cultured in EX media, CD3^+^γδTCR^+^Vγ9^+^ triple-positive γδ T cells were sorted by FACS, and the cytotoxicity and levels of secreted cytokines against OS cells were evaluated. OS cells were cultured with expanded γδ T cells at T/O ratios of 0.067:1–1:1 (expanded γδ T cells/OS cell ratios). Expanded γδ T cells showed increased cytotoxicity against the two OS cell lines in a ratio- and time-dependent manner. The cytotoxic effect of the expanded γδ T cells against KHOS/NP cells reached the highest level at 1:1 T/O ratios, and the viability of KHOS/NP cells decreased from 73.1% (24 h) to 23.1% (72 h) (Figure 3a).

Next, we evaluated cytokine levels in the supernatant media after culturing γδ T cells and KHOS/NP cells at T/O ratios of 0.25:1–1:1. Cytokine concentrations increased in a time- and ratio-dependent manner (Figure 3b). Compared with that in controls, cytokine concentrations were highest after 72 h of co-culture in EX media, with cytokine concentrations being 39.4 times (IFN-γ) and 18.7 times (TNF-α) higher at 1:1 T/O ratios. These results demonstrated that expanded γδ T cells have potent in vitro cytotoxicity against OS cells.

### 3.3. Addition of DOX/IFO to the γδ T-OS Co-Culture Condition

DOX and IFO are used in the standard chemotherapy for OS. DOX, an anthracycline antibiotics agent, works by slowing or stopping tumor cell growth. IFO, an alkylating agent, works by disrupting the tumor cell’s microtubule dynamics, and can also act as an immunosuppressive agent when used with adoptive immunotherapy [21,22].

The IC50 values of DOX and IFO were evaluated after 72 h of incubation. For KHOS/NP cells, the IC50 of DOX and IFO were 0.15 and 11,746 μM, respectively. For MG-63 cells, IC50 values were 0.14 and 10,690 μM (Figure 4). KHOS/NP and MG-63 OS cells were co-cultured with expanded γδ T cells. After 24 h, the cells were treated with DOX or IFO, and cell viability was serially assessed at 24, 48, and 72 h.

The addition of DOX tended to increase the cytotoxicity of expanded γδ T cells against the two OS cell lines in a dose-, ratio-, and time-dependent manner (Figure 5a,b and Table 1). Twenty-four hours after adding DOX, synergism was observed by CI analysis at ED50 (median effective dose) and ED75 in KHOS/NP cells (CI value: 0.453 and 0.983, respectively) and ED50 in MG-63 cells (CI value: 0.829). Additionally, the addition of DOX tended to increase the cytotoxicity after 48 and 72 h. However, the addition of IFO did not increase the cytotoxic effect of γδ T cells on OS cells.

## 4. Discussion

In this study, we successfully expanded γδ T cells from human PBMCs and established an optimal protocol. We evaluated various culture conditions and found that the addition of zoledronate (1 μM) and rhIL-2 (100 IU) yielded the highest number of CD3^+^γδTCR^+^Vγ9^+^ γδ T cells after 10-day culture. The concentrations of IFN-γ and TNF-α were the highest under these conditions, indicating that our protocol could promote efficient expansion and activation of γδ T cells. The expanded γδ T cells exhibited cytotoxicity against KHOS/NP and MG-63 cells in a ratio- and time-dependent manner. The addition of DOX enhanced the cytotoxic effect of expanded γδ T cells against the two OS cell lines, whereas IFO did not.

CD28 has diverse effects on T cells and acts as a major costimulatory receptor to promote the full activation of naïve CD4^+^ and CD8^+^ T cells [23,24]. Previous studies have explored the functional role of CD28 in conventional αβ T cells [25,26]. We investigated whether CD3/CD28 co-stimulation increases γδ T cell expansion and activation, including cytokine secretion and direct cytotoxicity. Counter-intuitively, the CD28 co-stimulatory molecules did not influence the expansion and activation of γδ T cells. Although the total PBMCs count significantly increased in EX28 media, CD3^+^ T cells (both CD3^+^αβTCR^+^ and CD3^+^γδTCR^+^ T cells) were barely observed. The proportion of CD25^+^ cells, a surrogate marker of T cell activation, was higher in EX28 media than in EX media. However, neither CD25^+^αβTCR^+^ nor CD25^+^γδTCR^+^ T cells increased after 10-day culture in each medium condition. Although further research is needed, we assumed that the increased number of CD25^+^ cells was because the αβ T cells that were not expanded to γδ T cells were activated and became CD25^+^ T cells (DN2 and/or DN3) by the CD3/CD28 activator. Moreover, the supernatant concentrations of IFN-γ and TNF-α increased in EX28 media, but this was not a significant increase, considering the increased cell counts. Our data indicate that CD3/CD28 did not exert any effect on the expansion and activation of human γδ T cells.

Ex vivo expanded γδ T cells showed a direct toxic effect on the two OS cells. Previous studies have revealed that γδ T cells produce cytokines and exert antiproliferative and preapoptotic effects on tumor cells. Similarly, we observed that the concentrations of IFN-γ and TNF-α in the supernatant (co-culture with expanded γδ T cells and KHOS/NP OS cells) increased in a ratio- and time-dependent manner and were directly cytotoxic to OS cells.

Several studies have demonstrated the efficacy of combination chemotherapy [27,28]. We evaluated whether the addition of DOX/IFO increased the cytotoxic effect of γδ T cells on OS cells. DOX and IFO are the standard chemotherapy regimens for patients with OS. We observed that DOX enhanced the expanded γδ T cell-mediated cytotoxic effects on the two OS cell lines, whereas IFO did not. The addition of DOX increased the cytotoxic effect of γδ T cells on OS cells, and CI analysis suggested synergism, especially after 24 h of incubation. However, the addition of IFO did not have any effect on OS cells. DOX is frequently used in combination with novel immunotherapeutic agents [29,30,31]. Previous studies have reported that DOX increases tumor immunogenicity and directly kills tumor cells. Our results suggest the possibility of a new treatment strategy for OS, such as the combination of DOX with γδ Τ cells.

The present study had several limitations. First, γδ T cells were expanded from PBMCs obtained from healthy volunteers and tested on commercially available OS cell lines; therefore, the cytotoxic effects of γδ T cells observed in our study might differ in vivo. Ideally, the cytotoxic effects of γδ T cells should be tested against OS cells obtained from the same patient. Second, the exact nature of the cell population expanded by CD3/CD28 agonists has not been thoroughly evaluated. Certainly, CD3/CD28 agonists do not influence the expansion of γδ T cells. However, the nature of how the number of CD25-positive cells increased with the addition of the CD3/CD28 agonist, which were neither γδ T cells nor αβ T cells, was elusive. Therefore, identification and characterization of the additional cells are needed. Finally, the synergistic mechanism of expanded γδ T cells and DOX was not identified because of the preliminary nature of our experiment. DOX is known to be an immunogenic chemotherapeutic agent and has been used in experiments exploring cancer immunotherapy. Further studies are necessary to elucidate the possible interactions between DOX and γδ T cells.

Taken together, we successfully expanded and activated γδ T cells from human PBMCs using IL-2 and zoledronate and observed that expanded γδ T cells had potent in vitro cytotoxicity against OS cells through the production of IFN-γ and TNF-α. The combination of DOX and γδ T cells showed synergistic cytotoxic effects on OS cells. Our data suggest that γδ T cells may enhance the effects of chemotherapeutic agents against OS and the possibility of a new treatment strategy, including chemo-immunotherapy, for OS.

## Figures and Tables

**Figure 1 cells-11-02164-f001:**
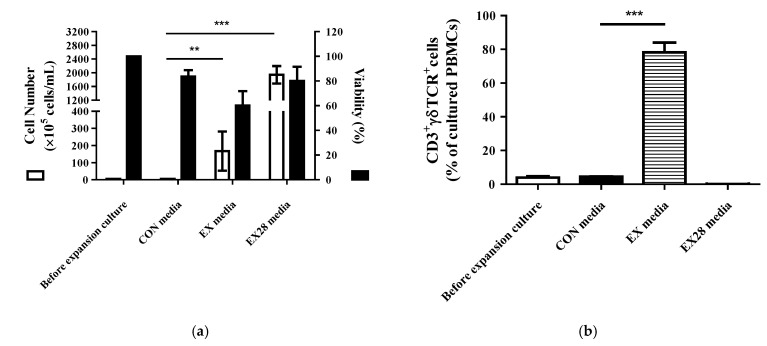
Optimization of human γδ T cell expansion. Peripheral blood mononuclear cells (PBMCs) from healthy donors were cultured in the indicated media. (**a**) Cell proliferation and viability assays. After expansion, the cell count and viability were assessed by trypan blue staining using an automatic cell counter. (**b**) Flow cytometric analysis of CD3^+^γδTCR^+^ cell populations. After expansion, single cells were stained and analyzed using the BD FACSCanto II and BD FACSVerse System. The percentages are represented as the mean ± SEM, and statistical significance was determined by comparison with the CON media group (** *p* < 0.01, *** *p* < 0.001).

**Figure 2 cells-11-02164-f002:**
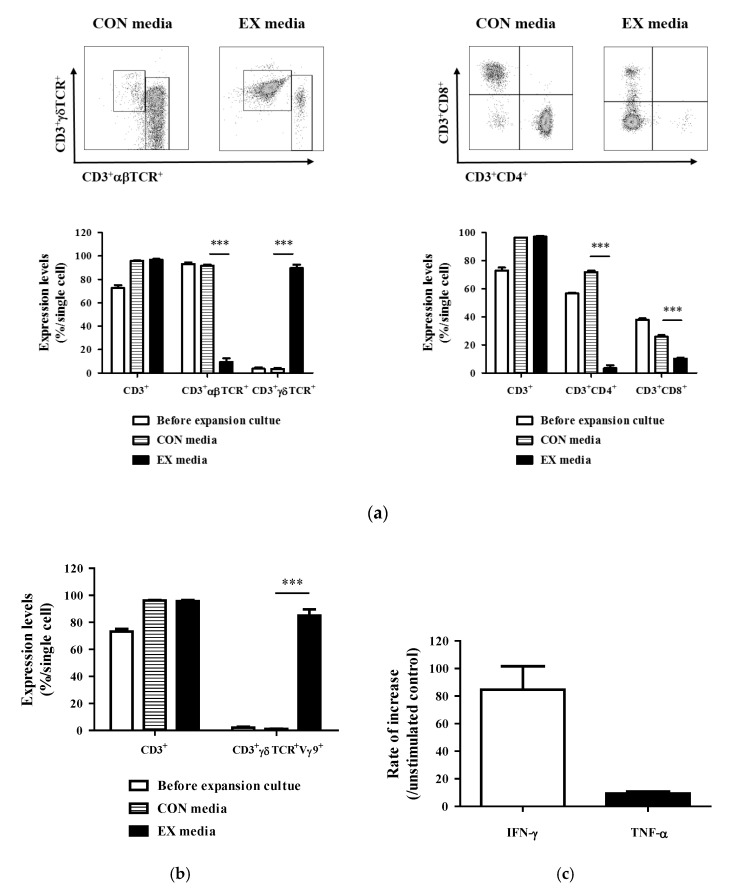
Activation and cytokine production by expanded γδ T cells. γδ T cell-derived peripheral blood mononuclear cells (PBMCs) were cultured under optimum culture conditions. (**a**,**b**) Flow cytometric analysis of changes in activation marker expression. After expansion, single cells were stained and analyzed using the BD FACSVerse System. (**c**) ELISA for cytokine production. After expansion culture, the supernatant concentrations of both IFN-γ and TNF-α were measured using an ELISA kit. The percentages are represented as the mean ± SEM, and statistical significance was determined by comparison with the CON media group (*** *p* < 0.001).

**Figure 3 cells-11-02164-f003:**
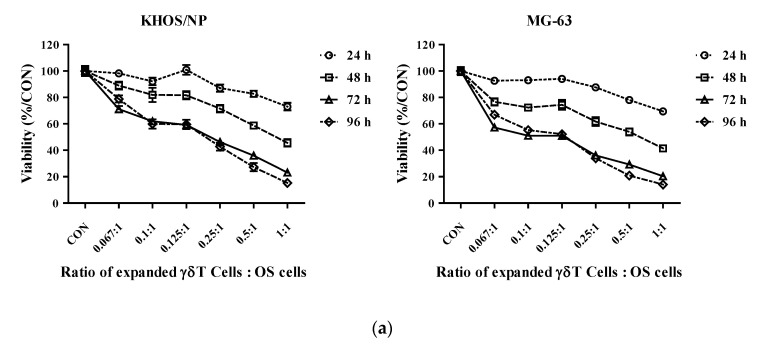
Cytotoxicity and cytokine production by expanded γδ T cells in osteosarcoma (OS) cells in vitro. The expanded γδ T cells were co-cultured with OS cells. (**a**) Cytotoxicity assays. After co-culture with OS cells, cytotoxicity and cell viability were measured using the EZ-CyTox cell viability assay kit after 24, 48, 72, and 96 h. (**b**) ELISA for cytokine production. After co-culture with KHPS/NP cells, the supernatant concentrations of both IFN-γ and TNF-α were measured using an ELISA kit. The percentages are represented as the mean ± SEM.

**Figure 4 cells-11-02164-f004:**
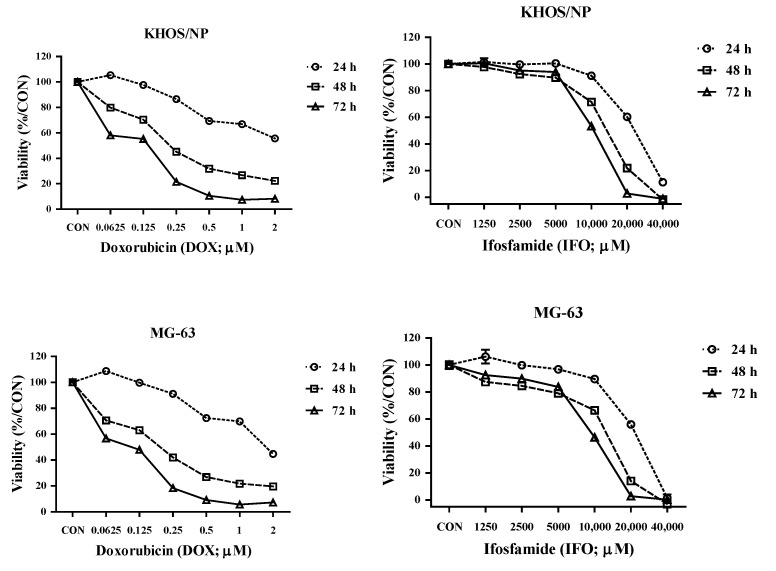
Cytotoxicity of anticancer drugs in osteosarcoma (OS) cells in vitro. KHOS/NP (**upper**
**panel**) and MG-63 (**lower panel**) OS cells were treated with doxorubicin (DOX) or ifosfamide (IFO), and cell viability and IC50 values were measured using the EZ-CyTox cell viability assay kit. The percentages are represented as the mean ± SEM.

**Figure 5 cells-11-02164-f005:**
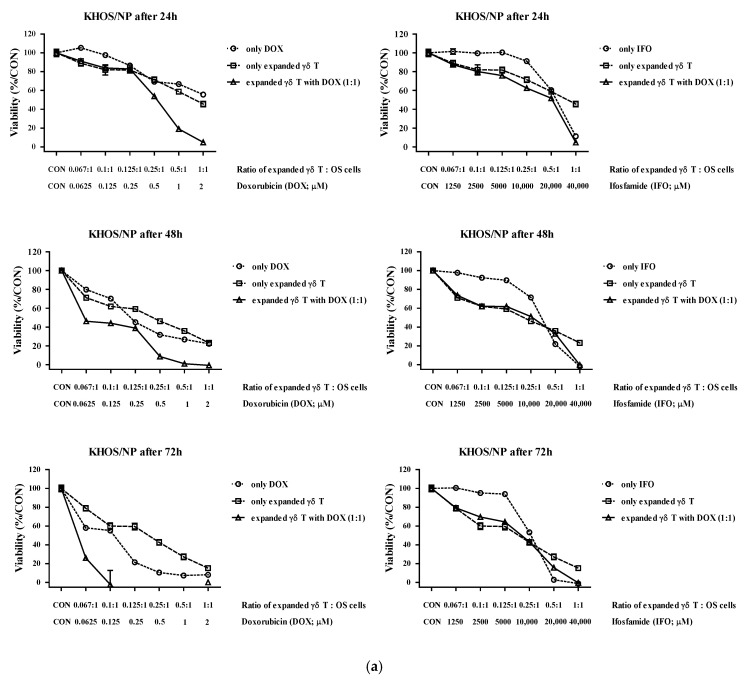
Combined cytotoxicity of anticancer drugs and expanded γδ T cells against osteosarcoma (OS) cells in vitro 24 h prior to drug treatment, KHOS/NP (**a**) and MG-63 (**b**). OS cells were pretreated with expanded γδ T cells. Cytotoxicity and cell viability were measured using the EZ-CyTox cell viability assay kit after 24, 48, and 72 h, and the percentages are represented as the mean ± SEM.

**Table 1 cells-11-02164-t001:** Combination Index (CI) values of the interaction between expanded γδ T cells with DOX against osteosarcoma (OS) cells.

	KHOS/NP Cells		MG-63 Cells	
	CI Values at	Degree of Additive /Synergism	CI Values at	Degree of Additive /Synergism
ED50	0.45283	synergism	0.82904	synergism
ED75	0.98269	synergism	2.31524	antagonism
ED90	2.22817	antagonism	7.63283	antagonism

KHOS/NP and MG-63 cells treated with different concentrations/ratio combinations of expanded γδ T cells and DOX were assayed for cell viability at 24 h, and CI values were calculated using CalcuSyn software. The concentrations/ratio used are as follows: DOX-treated (μM): 0.125, 0.25, 0.5, 1, 2; γδ T cell-treated (γδ T cells/OS cells ratio): 0.1, 0.125, 0.25, 0.5, 1; DOX and γδ T cell-treated: 0.125 DOX+0.1 γδ T cells, 0.25 DOX+0.125 γδ T cells, 0.5 DOX+0.25 γδ T cells, 1 DOX+0.5 γδ T cells, 2 DOX+1 γδ T cells; ED50, 75, 90: effective dose that kills 50, 75, and 90% of the cells.

## Data Availability

Not applicable.

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
