# Peer review of "Therapeutic Potential of Ex Vivo Expanded γδ T Cells against Osteosarcoma Cells"

_cells, 2022, doi:10.3390/cells11142164_

Round 1
Reviewer 1 Report
Comments to the authors,
Osteosarcoma is a malignant bone tumor, and treatment options are limited to patients. In this study, Ko et al. aimed to find better treatment for osteosarcoma and examined the efficacy of gdT cell-mediated antitumor activity in combination with conventional chemotherapeutic agents. Using human PBMCs, the authors first showed efficient expansion of PBMC-derived gfT cells with production of IFN-g and TNF-a in culture medium with recombinant IL-2 + zoledronate. Then, the authors showed that coculture of osteosarcoma cell lines with the gdT cells resulted in inhibition of osteosarcoma viability, suggesting potential cytotoxity of gdT cells against osteosarcoma. Finally, the authors showed increased cytotoxic effect of gdT cells in combination with chemotherapeutic agent Doxorubicin. Collectively, the authors concluded that combination of Doxorubicin with gdT cells may enhance the effects of chemotherapeutic agent against osteosarcoma.
Although this study has some limitation as the authors discussed, the results may have potential to contribute to this research field. However, I have couple of questions that revisions are required.
Main;
1. In the coculture experiments, the authors showed gdT cell-mediated cytotoxic effects by viability of osteosarcoma cell line cells. However, according to the materials and methods, the authors measured all viable cells in the culture, which means that the results did not indicate cell death of osteosarcoma cell line only. The authors have to use different method to measure cytotoxicity but not viability.
2. In the coculture experiments, the authors tested not only ratio of gdT cell: Osteosarcoma cells but also concentration of chemotherapeutic agent Doxorubicin/Ifosfaminde, and then mentioned that Doxorubicin tended to increase the cytotoxicity against osteosarcoma cell lines in a dose-, ratio- dependent manner. However, the results are misleading because both values (cell ratio and chemical dose) were varied. This experimental design cannot demonstrate cell ratio-dependency nor chemotherapeutic agent dose-dependency. When change the cell ratio, the dose of chemotherapeutic agent should be fixed, and vice versa.
Minor;
It is better to describe the background of chemotherapeutic agent Doxorubicin/Ifosfaminde briefly in the results section. Most of readers do no know what these chemicals are.
Author Response
Response to Reviewer 1 Comments
June 23, 2022
Dear Reviewer:
We would like to re-submit the attached manuscript entitled “Therapeutic potential of ex vivo-expanded gd T cells against osteosarcoma cells.” The manuscript ID is cells-1765840.
The manuscript has been carefully rechecked and appropriate changes have been made in accordance with the reviewers’ suggestions (highlighted in yellow in the revised manuscript). The responses to their comments have been prepared and attached herewith.
We thank you and the reviewers for the thoughtful suggestions and insights, which have enriched the manuscript and produced a more balanced and better account of the research. We hope that the revised manuscript is now suitable for publication in your journal.
Thank you for your consideration. We look forward to hearing from you.
Sincerely,
Yunmi Ko & Jun Ah Lee
Department of Pediatrics, Center for Pediatric Cancer, National Cancer Center
323 Ilsan-ro, Ilsandong-gu, Goyang-si, Gyeonggi-do, 10408, South Korea
+82-31-920-1604
junahlee@ncc.re.kr
Point 1: In the coculture experiments, the authors showed gdT cell-mediated cytotoxic effects by viability of osteosarcoma cell line cells. However, according to the materials and methods, the authors measured all viable cells in the culture, which means that the results did not indicate cell death of osteosarcoma cell line only. The authors have to use different method to measure cytotoxicity but not viability.
Response: We apologize for the confusion. After co-culture with expanded gd T cells and OS cells with or without doxorubicin (DOX) or ifosfamide (IFO), we removed old media with floating gd T cells and then measured the viability of adherent OS cells. We have revised the Materials and Methods as follows: “DOX (LC Laboratories, MA, USA) and IFO (Selleck Chemicals LLC, TX, USA) were diluted in culture medium, each with various concentrations: DOX, 0.0625–2 µM; and IFO, 1250–40,000 µM. After removing the old media containing gd T cells, the viability of adherent OS cells was evaluated using the EZ-CYTOX assay kit (DoGenBio, Korea) after 24, 48, and 72 h.”
Point 2: In the co-culture experiments, the authors tested not only ratio of gdT cell: Osteosarcoma cells but also concentration of chemotherapeutic agent Doxorubicin/Ifosfaminde, and then mentioned that Doxorubicin tended to increase the cytotoxicity against osteosarcoma cell lines in a dose-, ratio- dependent manner. However, the results are misleading because both values (cell ratio and chemical dose) were varied. This experimental design cannot demonstrate cell ratio-dependency nor chemotherapeutic agent dose-dependency. When change the cell ratio, the dose of chemotherapeutic agent should be fixed, and vice versa.
Response: We agree with your comment. Although the combination index (CI) analysis was performed using CalcuSyn software, the synergistic mechanism of expanded gd T cells and chemotherapeutic agent was not identified because of the preliminary nature of our experiment. We will attempt to establish an appropriate experimental design for comparing the drug combination synergism and report it in a future study.
Point 3: It is better to describe the background of chemotherapeutic agent Doxorubicin/Ifosfaminde briefly in the results section. Most of readers do not know what these chemicals are.
Response: Thank you for this suggestion. We agree with your comment and have incorporated the suggestion in the results section as follows: “DOX and IFO are used in the standard chemotherapy for OS. DOX, an anthracycline antibiotics agent, works by slowing or stopping tumor cell growth. IFO, an alkylating agent, works by disrupting the tumor cell’s microtubule dynamics, and can also act as an immunosuppressive agent when used with adoptive immunotherapy [21,22].
Reviewer 2 Report
Although the manuscript is an interesting in the effect of gamma-delta T cells on osteosarcoma therapy there are some serious problems. The authors should be addressing them.
1) In the present experiments, it is an important role of gamma-delta T cells in inhibition of osteosarcoma viability. How do the authors speculate the inhibition mechanism? Do you have idea of cell-cell contact or paracrine mechanism? The authors should discuss them.
2) Did the authors examine the effect of conditional medium derived from CD3+/ gamma-delta T cells+/Vgamma8+ cells on osteosarcoma viability?
3) Why do the authors speculate that the treatment with Ex28 increase the CD25+ T cells (DN2 and/or DN3), but not gamma-delta T cells?
4) The MG-63 usually have been used as the pre-osteoblasts cells derived from human osteosarcoma. Did you examine the other cell lines, such as Saous-2 etc. and/or commercial human primary osteosarcoma?
5) It is very unclear the labels in ratio of gamma-delta T cells to OS cells. The authors should represent the ratio in based OS cell as constant 1 and various of gamma-delta T cells
Author Response
Response to Reviewer 2 Comments
June 23, 2022
Dear Reviewer:
We would like to re-submit the attached manuscript entitled “Therapeutic potential of ex vivo-expanded gd T cells against osteosarcoma cells.” The manuscript ID is cells-1765840.
The manuscript has been carefully rechecked and appropriate changes have been made in accordance with the reviewers’ suggestions (highlighted in yellow in the revised manuscript). The responses to their comments have been prepared and attached herewith.
We thank you and the reviewers for the thoughtful suggestions and insights, which have enriched the manuscript and produced a more balanced and better account of the research. We hope that the revised manuscript is now suitable for publication in your journal.
Thank you for your consideration. We look forward to hearing from you.
Sincerely,
Yunmi Ko & Jun Ah Lee
Department of Pediatrics, Center for Pediatric Cancer, National Cancer Center
323 Ilsan-ro, Ilsandong-gu, Goyang-si, Gyeonggi-do, 10408, South Korea
+82-31-920-1604
junahlee@ncc.re.kr
Point 1: In the present experiments, it is an important role of gamma-delta T cells in inhibition of osteosarcoma viability. How do the authors speculate the inhibition mechanism? Do you have idea of cell-cell contact or paracrine mechanism? The authors should discuss them
Response: We focused on Vγ9Vδ2 T cells among human gd T cells in our experiment. One of their peculiar functional properties is the production of pro-inflammatory cytokines (IFN-γ and TNF-α), which exert anti-proliferative and apoptotic effects on tumor cells. In co-culture with expanded gd T cells and KHOS/NP cells, we observed decreased viability of KHOS/NP cells and increased concentrations of IFN-γ and TNF-α in a ratio- and time-dependent manner. Therefore, we inferred that cytotoxicity was induced through the pro-inflammatory cytokines produced by gd T cells. We agree with you that more studies on the inhibitory mechanism are needed, which will be reported in a future study.
Point 2: Did the authors examine the effect of conditional medium derived from CD3+/ gamma-delta T cells+/Vgamma8+ cells on osteosarcoma viability?
Response: Unfortunately, we did not examine the effect of conditional medium derived from CD3+/ gamma-delta T cells+/Vgamma9+ cells. We focused on Vγ9Vδ2 T cells among human gd T cells, which are able to inhibit cancer progression in various models in vitro and in vivo, and are endowed with peculiar functional properties, which make them very good candidates for immunotherapy. Therefore, we established an optimal protocol for the expansion and activation of gd T cells in human PBMCs. Additionally, we sorted only CD3+/ gamma-delta T cells+/Vgamma9+ cells and attempted to examine their therapeutic potential against OS cells.
Point 3: Why do the authors speculate that the treatment with Ex28 increase the CD25+ T cells (DN2 and/or DN3), but not gamma-delta T cells?
Response: CD28 is present on the surface of the majority of TCR αβ-bearing T lymphocytes. We used a CD3/CD28 activator, which was designed to activate and expand human T cells and consisted of soluble antibody complexes that bind CD3 and CD28 cell surface ligands. Previous studies have explored the functional role of CD28 in conventional αβ T cells; therefore, we attempted to investigate whether CD28 can increase the expansion and activation of gd T cells. The CD3/CD28 activator did not influence the expansion and activation of gd T cells; however, when activation of viable CD3+ T cells was assessed by CD25 expression using flow cytometry, the number of CD25+ T cells increased. We assumed that it was because the αβ T cells that were not expanded to gd T cells were activated and became CD25+ T cells (DN2 and/or DN3) by the CD3/CD28 activator; however, further research is needed to test our hypothesis.
Point 4: The MG-63 usually have been used as the pre-osteoblasts cells derived from human osteosarcoma. Did you examine the other cell lines, such as Saous-2 etc. and/or commercial human primary osteosarcoma?
Response: Due to the preliminary nature of our experiment, KHOS/NP and MG-63 cells were examined. We agree with your comment that experiments should be conducted using other cell lines. Additional experiments will be conducted using various OS cell lines in a future study.
Point 5: It is very unclear the labels in ratio of gamma-delta T cells to OS cells. The authors should represent the ratio in based OS cell as constant 1 and various of gamma-delta T cells
Response: We have incorporated your suggestion in the revised manuscript.
Round 2
Reviewer 1 Report
I have read the author’s reply, but I’m not quite sure about their reply to Point 2. What I pointed was that the experimental design in Figure 5 was wrong, and I don’t think that the experiment in Figure 5 demonstrated synergistic action of DOX on gdT cell-mediated cytotoxicity against osteosarcoma cell lines. I didn’t ask the mechanism underlying the synergistic action between DOX and gdT cells (again, I’m not sure whether there is a synergistic action between DOX and gdT cells). In general, to calculate combination index, 1) the value of concentrations of each drug (or ratio of gdT cell and OC cells?) alone to exert X% effect, and 2) the value of concentrations of drugs in combination to elicit the same effect. Which figures showed these values?
Author Response
Please see the attachment.
Thank you
Sincerely,
Yunmi Ko & Jun Ah Lee

Round 3
Reviewer 1 Report
The authors clarified my concerns. I'm satisfied.
Author Response
Dear reviewer
Thank you for giving us the opportunity to improve and revise our manuscript. We thank you again for the thoughtful suggestions and insights
Sincerely,
Yunmi Ko & Jun Ah Lee
Center for Pediatric Cancer, Department of Pediatrics, National Cancer Center